# Research on the Effects and Mechanism of Carbon Emission Trading on the Development of Green Economy in China

**Ling Wang [1], Zhiying Chen [2,*] and Zheheng Huang [1]**

1   Energy Development Research Institute of China Southern Power Grid, Guangzhou 510663, China
2   Guangzhou Academy of Social Science, Guangzhou 510410, China
*   Correspondence: chenzhiying@gz.gov.cn

**Abstract:** Based on the panel data from 2004 to 2018, this paper evaluates the effect of carbon emission trading policies launched in 2014 on the development of green economy in pilot areas by using synthetic control methods, and further studies the mechanism of the policies affecting green economy by using the mediating effects test. The results show that carbon emission trading has a significant effect on the green economy, which is affected by the economic base and location conditions. Technological innovation and the use of clean energy are the main mechanisms by which to promote green economy. Finally, some corresponding policy recommendations are put forward.

**Keywords:** carbon emission trading; green economy; synthetic control method; mediating effect



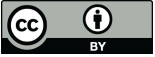

## 1. Introduction

In the past 30 years, environmental problems have been of great concern throughout the world. The universal question is how to minimize the damage of economic activities to the Earth's ecosystem and develop a "green economy" in which human society and the natural environment coexist harmoniously. Among these environmental problems, global climate change caused by greenhouse gas emissions is the most concerning issue. In 1992, the *United Nations Framework Convention on Climate Change* (UNFCC) was adopted and entered into force, which has laid the foundation for international cooperation to decrease greenhouse gas emissions. Subsequently, a series of international agreements, such as *Kyoto Protocol*, *Bali Roadmap*, *Copenhagen Climate Accord* and *The Paris Agreement*, were sequentially signed, and global agreement on the issue is that developed countries should be compelled to cut down emissions and developing countries should reduce emissions independently. In 2020, General Secretary Xi delivered a commitment at the 75th Session of The United Nations that China would strive to achieve the peak of carbon emissions by 2030 and carbon neutrality by 2060, which once again demonstrated China's determination to fulfill its emission reduction obligations and achieve green development.

Under the influence of the aforementioned series of international conventions, quite a large number of national and regional carbon trading markets have sprung up all over the world, such as European Union Emission Trading Scheme (EU ETS) and American Regional Greenhouse Gas Emission Reduction Action (RGGI), which guide market players to save energy and reduce emissions through market-oriented means. Since 2013, China has set out to experiment with seven carbon emission trading pilots in Beijing, Tianjin, Shanghai, Guangdong, Shenzhen, Hubei, and Chongqing, and after 2016, to be followed by two pilots in Fujian and Sichuan. On 16 July 2021, the national carbon emission trading market was officially launched, marking China's entrance into a new stage of green development. Therefore, it is pivotal to objectively and accurately evaluate the operational effects of the regional carbon emission trading pilots on the construction of the national carbon emission trading market and even the green development of China's economy in the future, which is the core content of this paper.

Based on this, this paper will evaluate the effect of carbon emission trading policies on the green economy in pilot areas, and test the path mechanism of the policies by the Synthetic Control Method (SCM) proposed by [1,2]. It consists of five sections: the first section is an introduction; the second section is a related literature research review; the third section is the design of research methods; the fourth section analyzes the test results of synthetic control; the fifth section analyzes the test results of the path mechanism of policies; and last are the summary and suggestions.

## 2. Related Literature Review

The concept of a "green economy" has been put forward for a long time, and can be traced back to [3]. Environmental economists believe that economic development must be sustainable for nature and human beings, and should not lead to social division or ecological crisis due to blind pursuit of production growth. They have advocated for the establishment of an "affordable economy" based on social and ecological conditions. Therefore, the essence of a green economy is a sustainable development economy with the coordinated development of ecology and economy at the core. It is an economic development mode characterized by maintaining the human living environment, reasonably protecting resources, and benefiting human health. A green economy is essentially consistent with a circular economy, which emphasizes resource conservation and recycling.

Obviously, with the increasingly severe conflict between economic development and environmental quality, the limitations of using GDP to measure the level of economic development have become increasingly obvious [4]. Therefore, how to accurately describe the impact of economic activities on the environment has become a widespread concern both at home and abroad. The current research has mainly formed two typical paradigms: one is called "Ecological Footprint" Rees [5], the connotation of which is a biologically productive area that maintains the survival of designated population units (a person, a region, a country, etc.) and absorbs waste emissions. The larger the area is, the greater the impact of human activities will have on the environment, and the more serious the ecological damage will become. Constanza et al. [6] put forward the Ecosystem Service Index (ESI) on this basis. In recent years, studies related to Ecological Footprint have emerged one after another, covering many fields, such as Environmental Kuznets Curve Bagliani et al. [7], real income Uddin et al. [8], environmental policies Solarin and Bello [9], FDI Udemba [10], etc. Now, it has become an important paradigm for assessing the degree of natural resource consumption and environmental damage.

Another kind of research paradigm is accounting for green GDP, that is, in the traditional GDP accounting system, the cost incurred by deducing ecological resource consumption and environmental damage. The most typical accounting method is the System of Environmental-Economic Accounting (SEEA) established by the United Nations Statistical Division (UNSD) in 1993, and the calculation formula is: "Green GDP = traditional GDP − consumption value of natural resources − loss value of environmental pollution − cost of environmental governance + incomes from environmental improvement." This is also the sole green GDP accounting method recognized by international authorities at present, and the latest version is SEEA-2012. Although the rationality of this accounting method is still controversial Peskin and Delos Angeles [11], it has been widely applied at home and abroad for the evaluation of the development level of green economy in a country or a region [12–17]. Thus, this paper will also use this method to calculate China's green GDP.

Although there are many studies on green economy at home and abroad [18–22], there are relatively few on how carbon emission trading policies affect the green economy. Currently, there are two main types of research on the effect of carbon emission trading policies at home and abroad. One type focuses on the impact of policies on carbon emissions-related variables, such as carbon emissions [23,24], carbon emission intensity [25,26], and carbon emission efficiency [27]. The other focuses on the impact of policies on traditional economic variables, such as output level [28] and energy structure [29,30], industrial

structure [31], etc. These studies evaluate the effect of carbon emission trading policies from different perspectives and provide reference for follow-up research, but there is still a lack of discussion of the overall impact on green economy. In fact, the fundamental goal of carbon emission trading in China is to promote the development of a green economy and realize the harmonious coexistence of economic construction and environmental protection. Therefore, there is a necessity to observe the impact of the implementation of policies on the development level of the green economy. Given the aforementioned factors, this paper studies the effects and mechanisms of the implementation of carbon emission trading policies on the development level of green economy in China, with China's carbon emission trading pilot as the object for observation.

In order to accurately evaluate the effect of carbon emission trading policies, this paper will use the synthetic control method proposed by [1,2]. In recent years, this method has been widely used to study international politics [32], environmental policies [33], energy policies [34], labor policies [35] and other fields. Similarly to the Difference-in-Difference Method (DID), which is commonly used in policy evaluation at present, the synthetic control method also divides the research objects into two categories: the Treatment Group, with implemented policies, and the Control Group, without implemented policies. Its uniqueness lies in fitting the weighted average of the Control Group to a "counterfactual" synthetic control object with similar characteristics to the Treatment Group. The difference between the real situation of the Treatment Group and the fitting situation of the synthetic control object is the effect of the policies. This method has four advantages: first, there is no extrapolation of the results, because the weight of the synthetic control method is non-negative and the sum is 1; second, the fitting results are transparent, and the difference between the Treatment Group and the counterfactual synthesis results is clearly visible; third, it is free from technical parameter search, and the weight of Control Group can be determined before policy intervention; and fourth, the counterfactual results are transparent, and the contribution of the Control Group to the counterfactual results is transparent as well. In particular, the synthetic control method allows avoidance of problems such as endogeneity and sample heterogeneity existing in the Difference-in-Difference Method [36], and thus it is known as "the most important policy evaluation innovation in the past 15 years" [37].

Finally, this paper will further explore the mechanism through which carbon emission trading policies affect green GDP. The current research mainly focuses on three possible mechanisms. The first mechanism is the adjustment of industrial structure: under the pressure of carbon emission trading policies, industrial enterprises with high pollution and high emissions will be phased out, and service enterprises with low pollution and low emissions will emerge in larger numbers [31,38]. The second mechanism is the adjustment of energy structure. As carbon emissions mainly come from the combustion of traditional fossil energy such as coal and oil, on the one hand, the carbon emission trading policies will force the consumption proportion of coal and oil to decrease, and replace it with the consumption proportion of natural gas with relatively low carbon emissions. On the other hand, it will also promote the transformation of traditional thermal power generation modes to clean power generation modes such as hydro-power, nuclear power, photo-voltaic energy, and wind energy [39]. The third mechanism is the improvement of the technical level. High-polluting enterprises will introduce cleaner production technology more voluntarily, improve production efficiency, and reduce waste of resources. The energy industry will also actively develop new energy technologies and reduce energy consumption [40,41]. These three mechanisms may be independent or interrelated. For instance, the improvement of the technical level may directly lead to the improvement of green GDP, it may lead to the improvement of green GDP by promoting the use of clean energy, or it may promote the adjustment of energy structure by promoting the adjustment of industrial structure. Finally, it may lead to the improvement of green GDP. Therefore, it is necessary to analyze the multi-chain intermediary effect of the policy transmission mechanism.

In summary, this paper will focus on the core issue of how carbon emission trading policies affect the green economy. Compared with the latest research, the innovations of this paper are as follows: firstly, the synthetic control method is used to evaluate the effect of carbon emission trading policies to ensure the credibility and robustness of the conclusions. Secondly, it analyzes the mechanism of carbon emission trading policies to promote a green economy, not only on the independent influence of different mechanisms, but also on the chain influence among various mechanisms. It also clarifies the real path for the policies to play their roles.

## 3. Research Design

### 3.1. Data Description

Considering the availability of data, this paper selects 29 provinces and cities, except Hainan and Tibet, as research samples. Because the Fujian and Sichuan pilot projects set out late and the transaction scale was smaller than that of the first batch of pilot projects, six of the first batch of pilot areas, including Beijing, Tianjin, Shanghai, Guangdong (including Guangdong and Shenzhen), Hubei, and Chongqing were selected as the Treatment Group, and the remaining 27 provinces were selected as the Control Group. The sample time interval is from 2004 to 2018, with 2004 as the base year for price deflation. Due to the fact that the first batch of pilot projects started in the second half of 2013, and the transaction volume in 2013 was very low, 2014 was selected as the starting year of the carbon emission rights policies. The data were gleaned from *China Statistical Yearbook*, *China Statistical Yearbook on Environment*, *China Energy Statistical Yearbook*, *China Land and Resources Statistical Yearbook*, WIND, etc.

### 3.2. Green GDP Accounting

In this paper, green GDP is calculated according to SEEA-2012, which is introduced by the United Nations Statistical Division. As an agent variable of the development level of green economy, the accounting formula is as follows:

Green GDP = traditional GDP − consumption value of natural resources − loss value of environmental pollution − cost of environmental governance + incomes from environmental improvement.

#### 3.2.1. Accounting of Consumption Value of Natural Resources

Water, energy, and cultivated land are selected as natural resources for the accounting of consumption value. The value of water resource consumption is calculated according to the formula "value of water resources = unit price of water resources × total amount of water resources consumption." At present, the most common international water resources price estimation method is as follows: $P = F/Q \times \alpha$, where $P$ stands for price of water resources; $F$ represents the total output value of water industry, which is replaced by regional GDP; $Q$ represents the total amount of water used; and $\alpha$ represents the willing payment coefficient of consumers in the water industry, which is calculated by formula (1) with reference to the research of [42], where $R$ represents the per capita water consumption (unit: cubic meters/person).

$$\alpha = \begin{cases} 3\%, & R \in [0,\ 500] \\ 3\% - (R - 500)/1250\%, & R \in (500, 3000) \\ 1\%, & R \in [3000, +\infty) \end{cases} \tag{1}$$

The energy consumption value is calculated according to the formula "value of energy consumption = unit price of energy × total energy consumption." Drawing lessons from the research of [14], the average price of standard coal in 2004 was CNY 1133 per ton as the unit price of energy.

The change value of cultivated land is calculated according to the formula "change value of cultivated land = unit price of cultivated land × change area of cultivated land." At present, the unit price of cultivated land is commonly estimated by the earnings method

of multiple markets. According to the relevant provisions of the *Law of Land Administration*, the average total agricultural output value of each region in the first three years is taken as the cultivated land output value of the region, then multiplied by 10 times to obtain the total cultivated land value. Finally, the unit price of cultivated land is obtained by dividing the total cultivated land value by the current cultivated land area stock.

### 3.2.2. Accounting for Loss Value of Environmental Pollution

In this paper, four main pollutants are selected for the accounting of loss value of environmental pollution: wastewater, waste gas, solid waste, and domestic garbage. The accounting formula is "loss value of environmental pollution = the cost of environmental degradation per unit of pollutant × total pollutant discharge." The cost of environmental degradation per unit of pollutant is obtained by dividing the total cost of environmental degradation by the total cost of environmental degradation generated by the pollutant, then dividing that by the total amount of pollutants discharged. According to the *2004 China Green National Economic Accounting Research Repor*t jointly issued by State Environmental Protection Administration and National Bureau of Statistics in 2004, the environmental degradation costs of the four types of pollutants are CNY 4.71 per ton, CNY 3605.29 per ton, CNY 7.16 per ton, and CNY 7.81 per ton, respectively.

### 3.2.3. Cost of Environmental Treatment

Drawing lessons from the research of [14], this paper takes the actual total investment in environmental pollution control in each region as equal to the cost of environmental treatment.

### 3.2.4. Income from Environmental Improvement

The types of income brought by environmental improvement covers a wide range, such as the recycling value of "three wastes" products and the extension of the lives of workers. In view of the availability of data, this paper mainly accounts for two types of income. The first is the economic value of carbon dioxide absorption by green space, which is calculated according to the formula "income from carbon sequestration of green space = stock of green space area × total carbon dioxide absorption per unit of green space × unit price of carbon dioxide." In this paper, the annual average trading price of carbon emission rights published by the European Climate Exchange is used as the unit price of carbon dioxide, and every hectare of green space can absorb about 328.5 tons of carbon dioxide every year [43]. Another kind of income is the economic value created by environmental improvement to prolong workers' lives, which is calculated according to the formula "income from life extension = environmental improvement degree × environmental influence coefficient on life × workers' remuneration in the current year." According to the research of [44], the influence coefficient of the environmental pollution index synthesized by the entropy weight method on workers' lifespans is about −0.23.

According to the above method, the accounting results of China's green GDP from 2004 to 2018 are shown in Tables 1 and 2, where Table 1 shows the national average green GDP as well as its internal structure from 2004 to 2018, and Table 2 shows the green GDP and its internal structure in 29 provinces in 2018. The data in Table 1 show that from 2004 to 2018, the national average green GDP increased from CNY 13.28 trillion to CNY 47.62 trillion, accounting for 78.92% to 86.06% of the traditional GDP. It can be seen that China's green development has achieved certain results, and the dependence on and loss of economic construction on the environment are gradually decreasing. From the internal structure of the green GDP, the consumption of natural resources accounts for the highest proportion of the traditional GDP, and energy consumption is the most important factor leading to the consumption of natural resources. At present, China's economic growth still mainly depends on the consumption of fossil energy, which is also an obstacle that China's high-quality economic development must overcome in the future. The proportion of environmental pollution losses caused by industrial "three wastes" and municipal garbage in the traditional GDP is decreasing year by year, due to the increasing yearly cost of

environmental treatment as well as the continuous optimization of industrial and energy structure. The proportion of income from environmental improvement to traditional GDP is low, and has a downward trend. First, the international carbon emission trading price has declined in recent years. Second, the increase in life expectancy has decreased, which leads to a decrease in total income. In some years, the negative income is mainly caused by the high pollution index and the decrease in life expectancy.

**Table 1.** National average green GDP and its internal structure unit from 2004 to 2018 (unit: CNY Trillion).

| Year | Green GDP | | Consumption of Natural Resources | | Loss of Environmental Pollution | | Cost of Environmental Treatment | | Incomes from Environmental Improvement | |
|---|---|---|---|---|---|---|---|---|---|---|
| | Total | Proportion | Total | Proportion | Total | Proportion | Total | Proportion | Total | Proportion |
| 2004 | 13.28 | 78.92% | 3.03 | 21.97% | 0.39 | 2.31% | 0.18 | 1.07% | 0.05 | 0.32% |
| 2005 | 15.03 | 78.96% | 3.44 | 22.06% | 0.42 | 2.23% | 0.22 | 1.15% | 0.08 | 0.41% |
| 2006 | 16.94 | 79.83% | 3.72 | 21.24% | 0.42 | 2.00% | 0.23 | 1.09% | 0.09 | 0.44% |
| 2007 | 18.78 | 79.73% | 4.11 | 21.11% | 0.42 | 1.80% | 0.27 | 1.13% | 0.02 | 0.09% |
| 2008 | 22.08 | 83.09% | 3.84 | 16.87% | 0.42 | 1.58% | 0.32 | 1.22% | 0.09 | 0.32% |
| 2009 | 23.73 | 81.32% | 4.66 | 19.01% | 0.40 | 1.38% | 0.36 | 1.24% | −0.02 | −0.08% |
| 2010 | 26.97 | 82.45% | 5.01 | 18.09% | 0.43 | 1.32% | 0.53 | 1.62% | 0.23 | 0.71% |
| 2011 | 29.60 | 82.30% | 5.37 | 17.54% | 0.55 | 1.52% | 0.53 | 1.46% | 0.07 | 0.20% |
| 2012 | 32.09 | 82.59% | 5.63 | 16.93% | 0.55 | 1.42% | 0.62 | 1.59% | 0.03 | 0.08% |
| 2013 | 33.92 | 81.64% | 6.40 | 18.22% | 0.55 | 1.32% | 0.70 | 1.68% | 0.02 | 0.05% |
| 2014 | 37.77 | 84.39% | 5.68 | 14.55% | 0.57 | 1.27% | 0.71 | 1.58% | −0.03 | −0.06% |
| 2015 | 38.16 | 84.68% | 5.72 | 14.55% | 0.56 | 1.24% | 0.68 | 1.50% | 0.05 | 0.12% |
| 2016 | 43.10 | 85.70% | 6.00 | 13.54% | 0.48 | 0.96% | 0.73 | 1.45% | 0.02 | 0.04% |
| 2017 | 45.03 | 85.66% | 6.40 | 13.86% | 0.46 | 0.87% | 0.71 | 1.36% | 0.03 | 0.06% |
| 2018 | 47.62 | 86.08% | 6.60 | 13.54% | 0.46 | 0.83% | 0.72 | 1.30% | 0.07 | 0.14% |

The data in Table 2 show that there are significant regional differences in the current development level of green economy in China. The green GDP of developed provinces and cities along the southeast coast, represented by Beijing, Tianjin, Shanghai, Guangdong, Jiangsu, and Zhejiang accounts for a high proportion of traditional GDP, exceeding 85%. In contrast, the green GDP of Shanxi, Gansu, Qinghai, Ningxia, and other provinces accounts for less than 70% of the traditional GDP, among which Ningxia has the lowest rate (57.6%), and the consumption of natural resources has reached 56.80%. It can be seen that the economic development of these provinces still comes at the expense of a large amount of natural resource consumption. This also shows that the development of green economy in different regions is unbalanced, although China has made some achievements in developing green economy. Therefore, it is necessary to take more active measures to deal with environmental degradation, which is one of the purposes of implementing carbon emission trading in China.

**Table 2.** Annual average green GDP and its internal structure of provinces and cities from 2004 to 2018 (unit: CNY 10 Billion).

| Province | Green GDP | | Consumption of Natural Resources | | Loss of Environmental Pollution | | Cost of Environmental Treatment | | Incomes from Environmental Improvement | |
|---|---|---|---|---|---|---|---|---|---|---|
| | Total | Proportion | Total | Proportion | Total | Proportion | Total | Proportion | Total | Proportion |
| Beijing | 102.88 | 87.49% | 11.41 | 10.75% | 0.72 | 0.61% | 2.75 | 2.34% | 0.17 | 0.14% |
| Tianjin | 65.95 | 85.36% | 9.98 | 14.83% | 0.51 | 0.66% | 0.89 | 1.15% | 0.06 | 0.08% |
| Shanghai | 118.07 | 86.52% | 16.17 | 13.44% | 1.25 | 0.92% | 1.26 | 0.92% | 0.29 | 0.21% |
| Guangdong | 331.16 | 87.91% | 39.19 | 11.61% | 4.37 | 1.16% | 2.80 | 0.74% | 0.81 | 0.22% |
| Hubei | 117.09 | 84.94% | 17.41 | 14.46% | 1.78 | 1.29% | 1.78 | 1.29% | 0.21 | 0.15% |
| Chongqing | 58.38 | 82.32% | 10.40 | 17.19% | 1.11 | 1.56% | 1.13 | 1.60% | 0.10 | 0.14% |
| Hebei | 118.96 | 74.80% | 34.84 | 28.06% | 2.59 | 1.63% | 2.92 | 1.84% | 0.29 | 0.18% |
| Shanxi | 46.42 | 66.44% | 20.14 | 40.51% | 1.76 | 2.52% | 1.78 | 2.55% | 0.24 | 0.34% |
| Inner Mongolia | 61.11 | 72.22% | 19.75 | 30.44% | 1.47 | 1.74% | 2.45 | 2.90% | 0.16 | 0.19% |
| Liaoning | 104.37 | 78.11% | 25.52 | 23.61% | 2.04 | 1.53% | 2.01 | 1.51% | 0.33 | 0.24% |
| Jilin | 56.84 | 83.24% | 10.01 | 17.18% | 0.90 | 1.32% | 0.66 | 0.96% | 0.12 | 0.18% |
| Heilongjiang | 70.41 | 88.23% | 7.35 | 10.15% | 1.16 | 1.46% | 1.09 | 1.37% | 0.21 | 0.27% |
| Jiangsu | 298.53 | 86.90% | 37.27 | 12.17% | 3.48 | 1.01% | 4.62 | 1.34% | 0.35 | 0.10% |
| Zhejiang | 191.98 | 86.17% | 25.77 | 13.08% | 2.32 | 1.04% | 2.83 | 1.27% | 0.11 | 0.05% |
| Anhui | 90.65 | 85.16% | 12.20 | 12.94% | 1.58 | 1.48% | 2.22 | 2.08% | 0.20 | 0.18% |
| Fujian | 109.34 | 86.60% | 14.27 | 12.74% | 1.48 | 1.17% | 1.27 | 1.00% | 0.10 | 0.08% |
| Jiangxi | 68.08 | 84.88% | 9.54 | 13.50% | 1.32 | 1.64% | 1.40 | 1.74% | 0.13 | 0.16% |
| Shandong | 262.27 | 83.12% | 45.53 | 16.86% | 3.25 | 1.03% | 4.76 | 1.51% | 0.28 | 0.09% |
| Henan | 154.79 | 83.00% | 27.28 | 17.14% | 2.73 | 1.46% | 1.93 | 1.04% | 0.25 | 0.13% |
| Hunan | 114.60 | 84.56% | 17.90 | 15.21% | 1.94 | 1.43% | 1.33 | 0.98% | 0.24 | 0.17% |
| Guangxi | 66.36 | 83.62% | 10.28 | 14.87% | 1.73 | 2.18% | 1.18 | 1.48% | 0.18 | 0.23% |
| Sichuan | 120.81 | 82.69% | 21.99 | 17.72% | 2.09 | 1.43% | 1.40 | 0.96% | 0.19 | 0.13% |
| Guizhou | 34.70 | 75.37% | 9.74 | 26.82% | 1.03 | 2.23% | 0.66 | 1.43% | 0.08 | 0.18% |
| Yunnan | 52.87 | 81.14% | 10.44 | 19.07% | 1.08 | 1.65% | 0.81 | 1.25% | 0.04 | 0.06% |
| Shanxi | 70.36 | 81.98% | 13.07 | 17.96% | 1.21 | 1.41% | 1.38 | 1.61% | 0.19 | 0.22% |
| Gansu | 25.35 | 74.89% | 7.27 | 27.38% | 0.63 | 1.87% | 0.63 | 1.85% | 0.03 | 0.10% |
| Qinghai | 7.01 | 62.27% | 3.79 | 50.68% | 0.29 | 2.53% | 0.20 | 1.75% | 0.02 | 0.19% |
| Ningxia | 7.96 | 57.60% | 5.01 | 56.80% | 0.44 | 3.18% | 0.43 | 3.12% | 0.02 | 0.12% |
| Xinjiang | 33.34 | 72.18% | 10.46 | 29.29% | 0.99 | 2.14% | 1.49 | 3.22% | 0.09 | 0.19% |

### 3.3. Synthetic Control Method

Suppose there are $N + 1$ regions, one of which is set up as a carbon emission trading pilot in the period $T_0$, while the other $N$ regions are not set up as carbon emission trading pilots. If $Y_{it}^T$ represents the green GDP of region $i$ after implementing carbon emission trading in $t$ period, and $Y_{it}^C$ represents green GDP of region $i$ without implementing carbon emission trading in $t$ period, then the green GDP which is actually observed by region $i$ in $t$ period can be expressed as $Y_{it} = Y_{it}^C + D_{it}\pi_{it}$, with $D_{it}$ as a dummy variable. If region $i$ is a carbon emission trading pilot in $t$ period, its value is 1, otherwise it is 0. $\pi_{it}$, which is also the research object of this paper, represents green GDP promotion of the carbon emission trading pilot in $t$ period to the region $i$. It is not difficult to see

that $\pi_{it} = Y_{it} - Y_{it}^C = Y_{it}^T - Y_{it}^C$. $Y_{it}^T$ is actually observed and $Y_{it}^C$, namely, the green GDP of the pilot areas without the implementation of carbon emission trading, needs to be estimated. This counterfactual result can be estimated by the following model [2]:

$$Y_{it}^C = \delta_t + \theta_t Z_i + \lambda_t \mu_i + \varepsilon_{it} \tag{2}$$

In Equation (2), $\delta_t$ is the time-fixed effect. $Z_i$ is an observable $(K \times 1)$ dimension control variable that is not affected by the carbon emission trading pilot. $\theta_t$ is an unknown parameter vector of the $(1 \times K)$ dimension. $\lambda_t$ is a common factor vector that cannot be observed in the $(1 \times F)$ dimension. $\mu_i$ is a regional fixation effect that cannot be observed in the $(F \times 1)$ dimension. $\varepsilon_{it}$ is a short-term shock that cannot be observed in various regions, and meets the average value of 0 at the regional level.

In order to determine $Y_{it}^C$, we can consider a weight vector $W = (w_1, w_2, \ldots, w_{i+1} \ldots w_{N+1})$ of the $(N \times 1)$ dimension, which satisfies $w_j \geq 0, j \neq i, \sum_{j \neq i} w_j = 1$. Each specific value of the vector $W$ represents a synthesis of a Control Group region to the carbon emission pilot region $i$, that is, $w_j$ is the weight corresponding to the Control Group region $j$. Therefore, the result variables of composite control are:

$$\sum_{j \neq i} w_j Y_{jt} = \delta_t + \theta_t \sum_{j \neq i} w_j Z_j + \lambda_t \sum_{j \neq i} w_j \mu_j + \sum_{j \neq i} w_j \varepsilon_{jt} \tag{3}$$

Suppose $(w_1^*, w_2^*, \ldots, w_{i-1}^*, w_{i+1}^* \ldots w_{N+1}^*)$ exists, so that:

$$\sum_{j \neq i} w_j^* Y_{j1} = Y_{i1}, \sum_{j \neq i} w_j^* Y_{j2} = Y_{i2}, \ldots, \sum_{j \neq i} w_j^* Y_{jT_0} = Y_{iT_0}, \sum_{j \neq i} w_j^* Z_j = Z_i \tag{4}$$

$\sum_{t=1}^{T_0} \lambda_t' \lambda_t$ is a non-singular matrix:

$$Y_{it}^C - \sum_{j \neq i} w_j^* Y_{jt} = \sum_{j \neq i} w_j^* \sum_{s=1}^{T_0} \lambda_t \left( \sum_{k=1}^{T_0} \lambda_t' \lambda_t \right)^{-1} \lambda_s' (\varepsilon_{js} - \varepsilon_{is}) - \sum_{j \neq i} w_j^* (\varepsilon_{jt} - \varepsilon_{it}) \tag{5}$$

Abadie et al. [2] proved that under general conditions, if the time before policies is longer than the time after policies, Equation (5) tends to equal 0. Therefore, in the policies period of $t \in [T_0 + 1, T]$, the counterfactual results of carbon emission rights pilot region $i$ can be expressed by the composite Control Group, namely, $\hat{Y}_{it}^C = \sum_{j \neq i} w_j^* Y_{jt}$. Therefore, the unbiased estimation of the policies' effect is:

$$\pi_{it} = Y_{it} - \hat{Y}_{it}^C = Y_{it} - \sum_{j \neq i} w_j^* Y_{jt}, t \in [T_0 + 1, T] \tag{6}$$

*3.4. Mediating Effect Test*

In order to further discuss the promotion mechanism of carbon emission trading policies on green GDP, referring to the multi-intermediary effect test method of [45], the following multi-intermediary model is made and the relevant path is shown in Figure 1:

$$Ggdp_{it} = cPilot_i \times Year_t + \eta_1 Control_{it} + \omega_{it} \tag{7}$$

$$Rd_{it} = a_1 Pilot_i \times Year_t + \sigma_{1it} \tag{8}$$

$$Indus_{it} = a_2 Pilot_i \times Year_t + d_{21} R\&d_{it} + \sigma_{1it} \tag{9}$$

$$Energy_{it} = a_3 Pilot_i \times Year_t + d_{31} R\&d_{it} + d_{32} Indus_{it} + \sigma_{2it} \tag{10}$$

$$Ggdp_{it} = c' Pilot_i \times Year_t + b_1 R\&d_{it} + b_2 Indus_{it} + b_3 Energy_{it} + \eta_2 Control_{it} + v_{it} \tag{11}$$

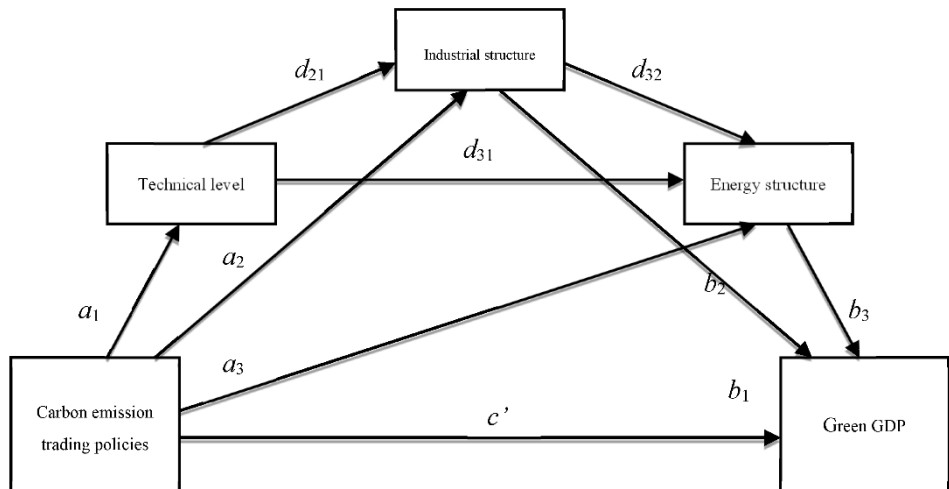

**Figure 1.** Impact mechanism of carbon emission trading on green GDP.

Among them, *R&D* represents the technical level, and the accounting method is the ratio of *R&D* funds to main business incomes from industrial enterprises above designated size. This reflects the increase in investment of high-emission enterprises in improving cleaner production technology caused by carbon emission trading policies. *Indus* represents the industrial structure, and the accounting method is the ratio of the added value of the secondary industry to the added value of the tertiary industry. This reflects the transfer of the secondary industry to the tertiary industry caused by the carbon emission trading policies. *Energy* represents the energy structure, and the accounting method is the ratio of electricity to total energy consumption (coal, oil, natural gas, and electricity). This reflects the transfer of fossil energy consumption to clean energy consumption caused by carbon emission trading policies. $c$ represents the total effect of the carbon emission trading policies on green GDP, and $c'$ represents the direct effect of the carbon emission trading policies on green GDP. The mediating effects include independent mediating effects $a_1b_1$, $a_2b_2$, and $a_3b_3$, as well as chain mediating effects $a_1d_{21}b_2$, $a_1d_{31}b_3$, $a_2d_{32}b_2$, $a_1d_{21}$, and $d_{32}b_3$, all tested by the bootstrap method [46].

### 3.5. Control Variables

In order to improve the robustness of the empirical results, this paper refers to the research of [31], and selects the following three factors that may affect green GDP as control variables:

(1) Urbanization rate: the impact of urbanization rate on green GDP is variable, and includes water resources consumption, energy consumption, cultivated land area change, green space area change, domestic garbage emission, and so on. In this paper, the ratio of urban population to resident population is used to calculate the urbanization rate.

(2) Government investment in environmental protection: the impact of government investment in environmental protection on green GDP comes from two aspects: on the one hand, it can reduce pollution emissions and improve environmental quality; on the other hand, it can lead to a reduction in other public services and reduce the total social output. In this paper, the ratio of government energy conservation and environmental protection expenditure to general budget expenditure is used to calculate government environmental protection investment.

(3) The number of medical staff per capita: the influence of the number of medical staff per capita on green GDP mainly lies in prolonging the average life of workers, thus increasing the income from environmental improvement. In this paper, the number of medical staff per thousand people is used for to calculate this.

## 4. Analysis of the Policies' Effect

### 4.1. Results of Synthesis Control

First of all, this paper constructs synthetic control provinces for each carbon emission trading pilot. Because Guangzhou and Shenzhen carbon emission trading pilots belong to Guangdong Province, this paper does not distinguish between them, but takes Guangdong Province as the pilot area. The selection criterion of the synthetic Control Group is minimization of the mean square error of the green GDP between the pilot areas and the synthetic control provinces in the period before the start of the carbon emission trading mechanism. The results are shown in Figure 2, in which the solid line represents the actual green GDP of the pilot areas and the dotted line represents the green GDP synthesized by the non-pilot provinces. It can be seen that the fitting effect of Beijing, Tianjin, Hubei, and Chongqing is poor, and the fitting effect of Shanghai and Guangdong is also poor. The reason for this is that the green GDP of Shanghai and Guangdong, which are, essentially, in the top two positions in China, is relatively high. Thus, it is difficult to carry out weighted fitting by other provinces. However, this will not affect the next analysis of this paper. Even if there are special circumstances that lead to an inability to find suitable synthetic control objects in the areas of policy implementation, it is also possible to explain that the policies are effective to a certain extent as long as it can be proved that certain policies have a significant impact on some countries or regions [47]. Therefore, the following analysis will focus on the Beijing, Tianjin, Hubei, and Chongqing pilot areas with good, fitting results.

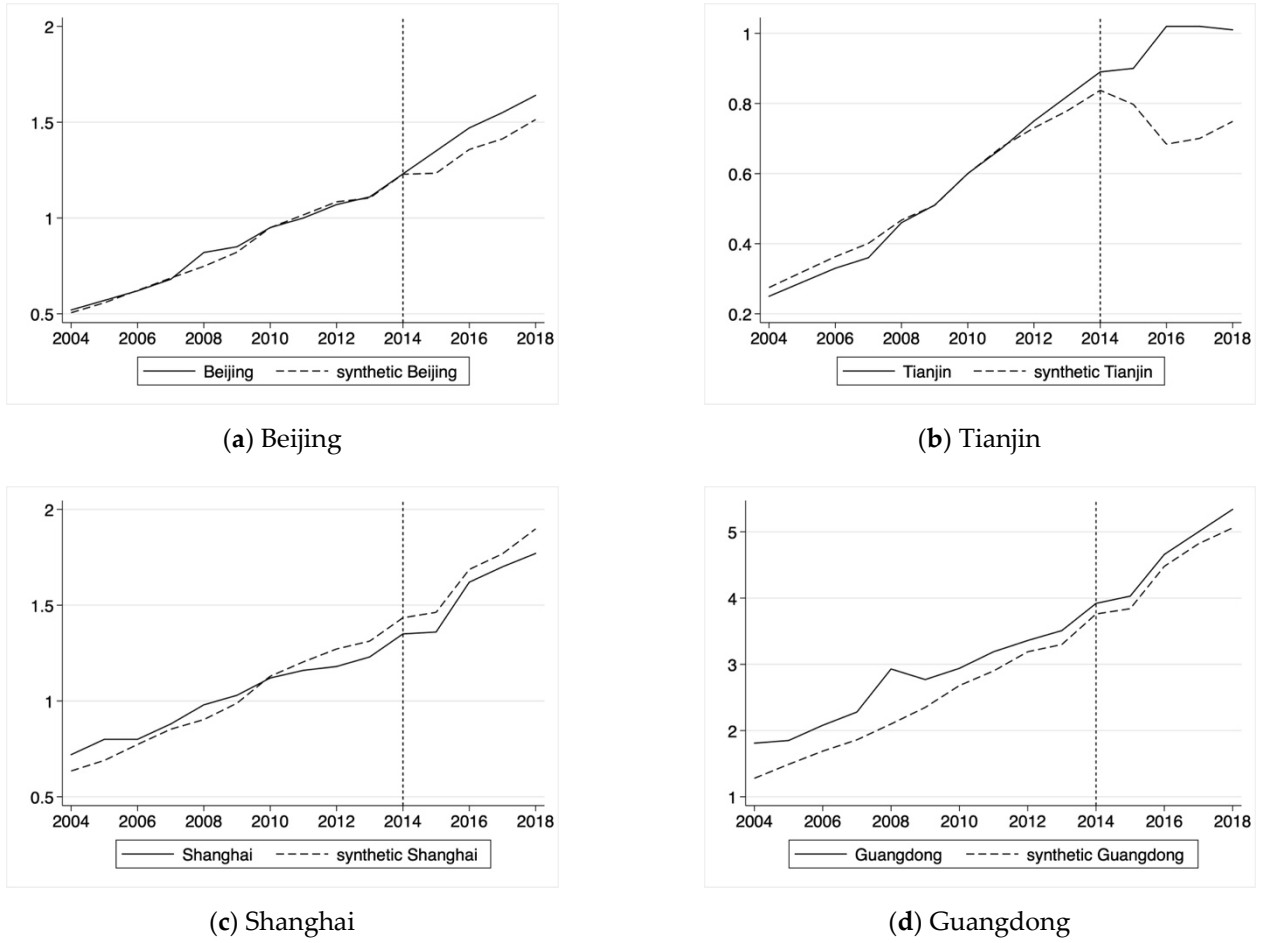

(**a**) Beijing

(**b**) Tianjin

(**c**) Shanghai

(**d**) Guangdong

**Figure 2.** *Cont.*

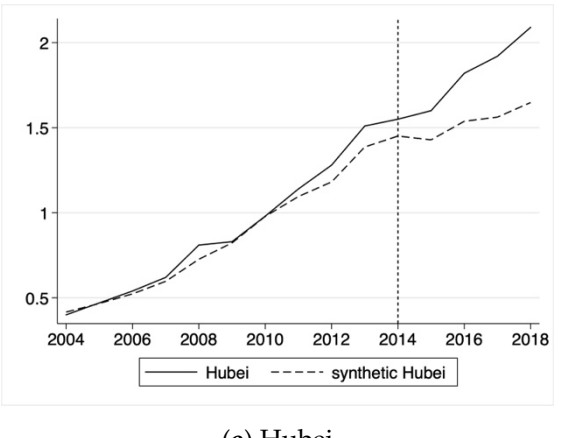

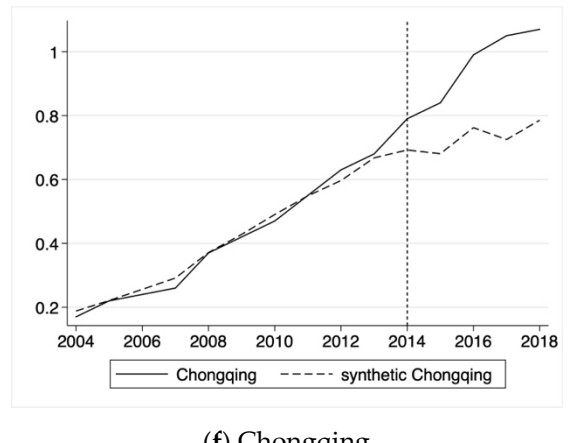

(**e**) Hubei                                      (**f**) Chongqing

**Figure 2.** Green GDP results of synthesis control of each carbon emission trading pilot.

Figure 2 shows that the actual green GDP of Beijing, Tianjin, Hubei, and Chongqing is basically consistent with the synthetic green GDP before the carbon emission trading pilot is launched, and the fit degree is very high. After the carbon emission trading pilot began, the actual green GDP and synthetic green GDP began to deviate, the former remaining above the latter, and with the passage of time, the gap between them widened. This shows that the launch of the carbon emission trading pilot has improved the green GDP of these four pilot areas, and this improvement effect has been continuously improved with the operation of the carbon emission trading pilot. It is worth noting that the separation of actual green GDP and synthetic green GDP in some pilot areas gradually appeared in 2011. It can be understood that the National Development and Reform Commission announced this policy as early as the end of 2011. Therefore, the pilot areas have started preparatory work, which would inevitably lead to the adjustment of local industrial structure, a reduction in energy consumption and pollution emissions, and promotion of the improvement of green GDP.

Table 3 summarizes the increase amount and range of actual green GDP compared with synthetic green GDP in Beijing, Tianjin, Hubei, and Chongqing since 2014. Vertically, from 2014 to 2018, the policies' stimulus effect of carbon emission trading on green economy gradually increased starting in 2014, and reached its peak in 2016. Then, the stimulus effect decreased, but it still significantly improved the green GDP of each pilot. Horizontally, the improvement rates of Tianjin and Chongqing is relatively high, while the improvement rate of Beijing is relatively low. It is not difficult to understand from Table 2 that Beijing's green GDP accounts for a high proportion, and the policies' stimulus effect is limited; Tianjin and Chongqing, which are mainly industrial and have a high proportion of consumption of natural resources, can fully demonstrate the stimulating effect of policies.

**Table 3.** Effects of pilot policies in Beijing, Tianjin, Hubei, and Chongqing (units: trillion CNY; %).

| Green GDP | Beijing | | Tianjin | | Hubei | | Chongqing | |
|---|---|---|---|---|---|---|---|---|
| | Ascend Quantity | Ascend Amplitude | Ascend Quantity | Ascend Amplitude | Ascend Quantity | Ascend Amplitude | Ascend Quantity | Ascend Amplitude |
| 2014 | 0.002 | 0.14 | 0.05 | 6.27 | 0.10 | 6.84 | 0.10 | 23.41 |
| 2015 | 0.12 | 9.43 | 0.10 | 12.86 | 0.17 | 12.00 | 0.16 | 29.89 |
| 2016 | 0.11 | 8.29 | 0.34 | 49.14 | 0.28 | 18.35 | 0.23 | 44.82 |
| 2017 | 0.14 | 9.71 | 0.32 | 45.64 | 0.36 | 22.90 | 0.32 | 36.22 |
| 2018 | 0.13 | 8.35 | 0.26 | 34.88 | 0.44 | 26.81 | 0.28 | 23.41 |

*4.2. Validity Test*

4.2.1. Placebo Test

The above results show that the actual green GDPs of Beijing, Tianjin, Hubei, and Chongqing are significantly higher than the counterfactual green GDP after 2014. However, whether the main cause of this difference is the initiation of a carbon emission trading mechanism or other unobserved exogenous factors still requires further verification. Therefore, this paper uses the placebo test proposed by [2] to test the effectiveness of the policies. The basic notion is that one assumes that a province which has not been set up as a carbon emission trading pilot during the sample period is affected by the same policies as the pilot provinces. The same synthetic control method is used to test the effect of the policies. If the policies' effect is not obvious, it shows that the promotion effect of the green GDP in pilot provinces does come from carbon emission trading, and the above empirical conclusion is effective; otherwise, it indicates that the promotion effect of green GDP in pilot provinces does not come from carbon emission trading, and the above empirical conclusion is invalid. Drawing lessons from the research of [31], this paper selects the four provinces with the highest weight in synthesizing four pilot green GDPs for the placebo test, namely Ningxia (accounting for 44.4% of synthetic Beijing), Xinjiang (accounting for 56.6% of synthetic Tianjin), Hunan (accounting for 57.1% of synthetic Hubei), and Inner Mongolia (accounting for 58.3% of synthetic Chongqing). The results are shown in Figure 3. It can be seen that the difference between the real GDP and the green GDP in Ningxia is small. The actual green GDP of Xinjiang and Inner Mongolia is significantly lower than the synthetic green GDP after 2014. The actual green GDP of Hunan is slightly higher than the synthetic green GDP, but the difference is also small. It can be seen that the substantial increase in green GDP in Beijing, Tianjin, Hubei, and Chongqing after 2014 is not due to accidental factors, but due to the initiation of carbon emission trading mechanisms.

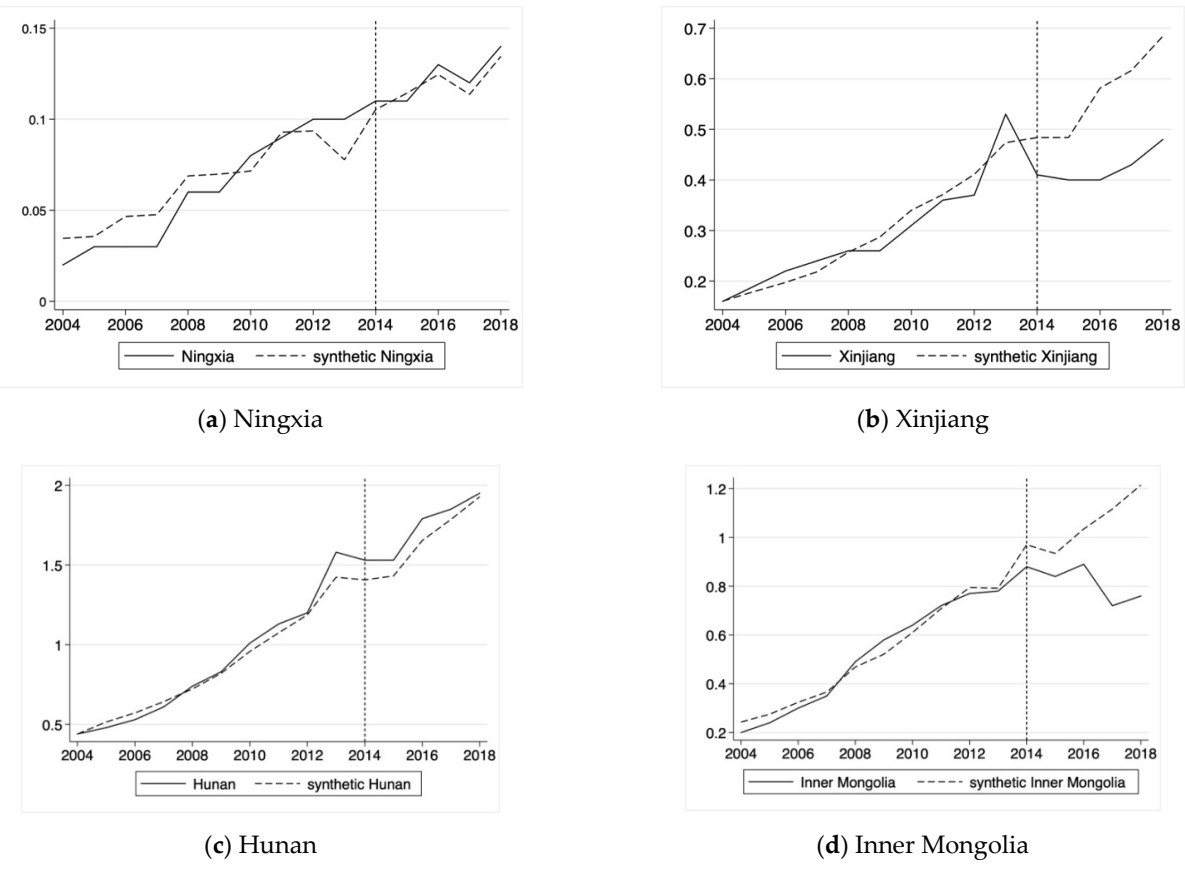

(**a**) Ningxia

(**b**) Xinjiang

(**c**) Hunan

(**d**) Inner Mongolia

**Figure 3.** Results of the placebo test.

4.2.2. Sorting Test

Although the results of the placebo test verify that carbon emission trading can promote the green GDP in pilot areas, the difference between the actual green GDP and the synthetic green GDP may still be the result of superposition of various policies factors. Thus, it is necessary to test the statistical significance of the policies' effects. Abadie et al. (2010) adopt a permutation test method. Its method is to assume that all Control Groups will start carbon emission trading from 2014, and to then adopt the synthetic control method to construct the synthetic control objects of these Control Groups, estimate the implementation effect of the policies under the above assumptions, and compare the policies' effect with the actual pilots. If the gap in the policies' effect between the two is not obvious, it shows that the effect of the promotion of carbon emission trading on green GDP is not statistically significant. On the contrary, it is reasonable to believe that this promotion is significant. In this paper, the objects used in the sorting test include the amount and range of green GDP increase. It should be pointed out that the ranking test requires that the composite control results of the Control Group before the implementation of the policies have a high degree of fitting, that is, the Root Mean Square Prediction Error (RMSPE) should not be too large. The reason is obvious: if the fit degree prior to the implementation of the policies is low, then the large gap between reality and the composite control result after the implementation of the policies may still be caused by the low fit degree rather than any significant effect of the policies. Therefore, referring to the research of [2], this paper excludes Jiangsu, Guizhou, Qinghai, and Ningxia, whose RMSPE exceeded the maximum value of Beijing, Tianjin, Hubei, and Chongqing before 2014, as well as 19 provinces engaged with the ranking test. The results are shown in Figures 4 and 5, in which black solid lines represent the results of Beijing, Tianjin, Hubei and Chongqing, and dotted lines represent the results of other Control Group provinces.

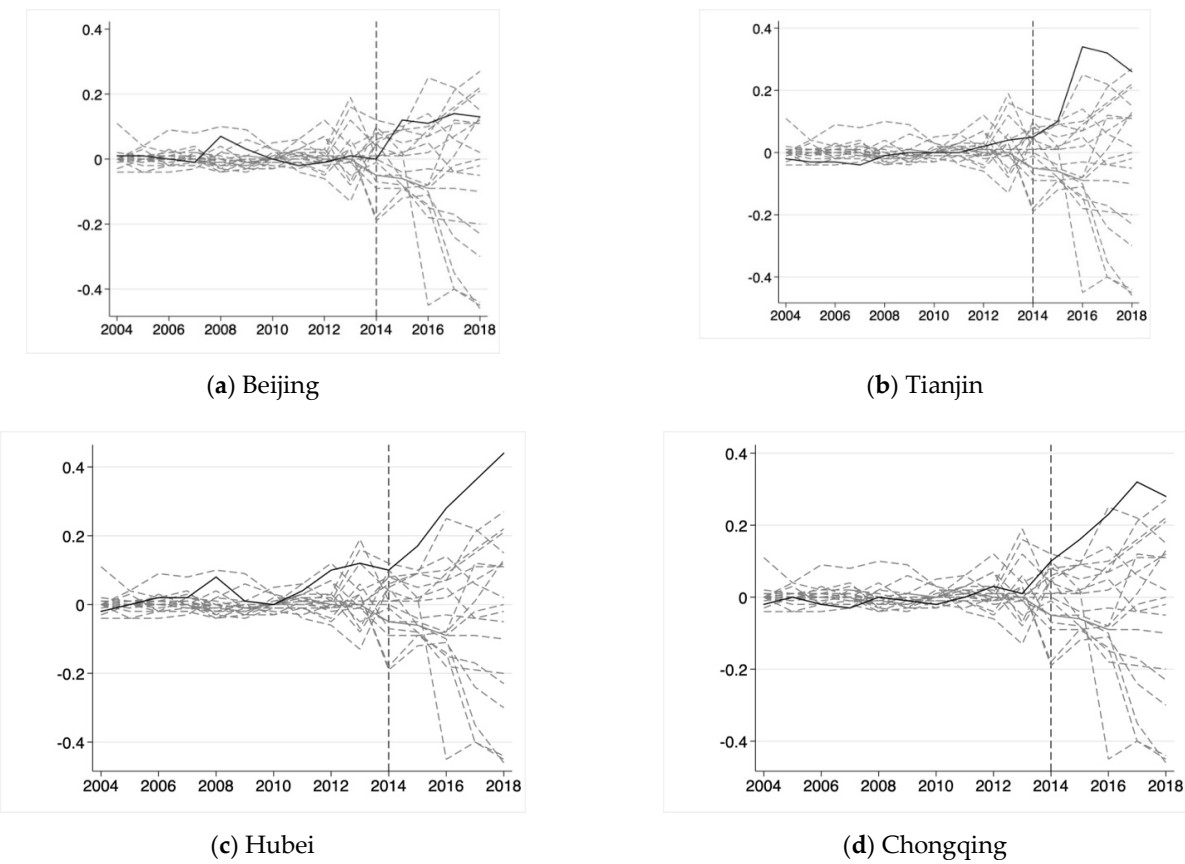

**Figure 4.** Results of sorting test—increase in green GDP.

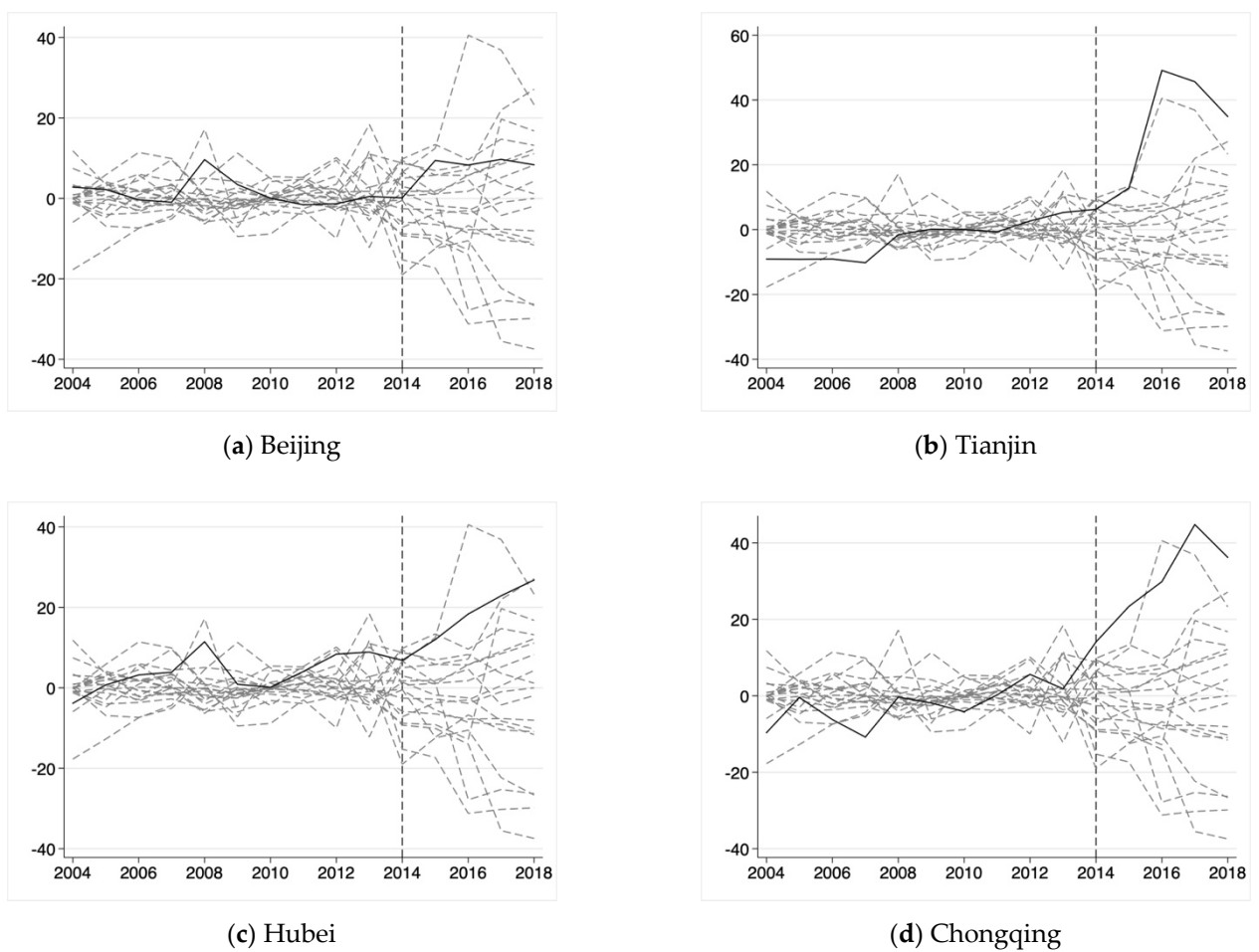

(**a**) Beijing

(**b**) Tianjin

(**c**) Hubei

(**d**) Chongqing

**Figure 5.** Results of sorting test—the increasing rate of green GDP.

It can be seen that, before 2014, there is little difference between the actual green GDP and the synthetic green GDP in most provinces. Since 2014, there have been differences between them. From the perspective of the increase in green GDP, Beijing is lower than four provinces, while Tianjin, Hubei, and Chongqing are higher than all of the other provinces. Therefore, it can be considered that the carbon emission trading policies have significantly increased the green GDP of Beijing by 10%, while the green GDP of Tianjin, Hubei, and Chongqing is significantly increased by 5%. From the perspective of the promotion of green GDP, Beijing is lower than six provinces, Hubei is lower than one province, and Tianjin and Chongqing are higher than all other provinces. Therefore, it can be considered that the carbon emission trading policies have significantly promoted the green GDP of Beijing and Hubei by 10%, and that of Tianjin and Chongqing by 5%. Generally speaking, the promotion effect of carbon emission trading policies on green GDP is relatively low in Beijing, but very significant in Tianjin, Hubei, and Chongqing.

*4.3. Robustness Test*

The above verifies the effectiveness of carbon emission trading policies, but whether choosing Control Groups of different weights will have an impact on the effectiveness of the policies still needs further verification. Therefore, the robustness test will be carried out by the iterative method below. Specifically, aiming at Beijing, Tianjin, Hubei, and Chongqing, a synthetic group province with positive contribution is deleted in turn by iteration in order to test whether the promotion effect of carbon emission trading policies on green GDP is affected by the weight of a synthetic Control Group. The results are shown in Figure 6. It can be seen that the weight reorganization and merger of the changed Control Groups

in the four pilot areas do not change the above conclusion, and the actual green GDP is significantly higher than the synthetic control green GDP after repeated iterative screening. Thus, the conclusion is robust.

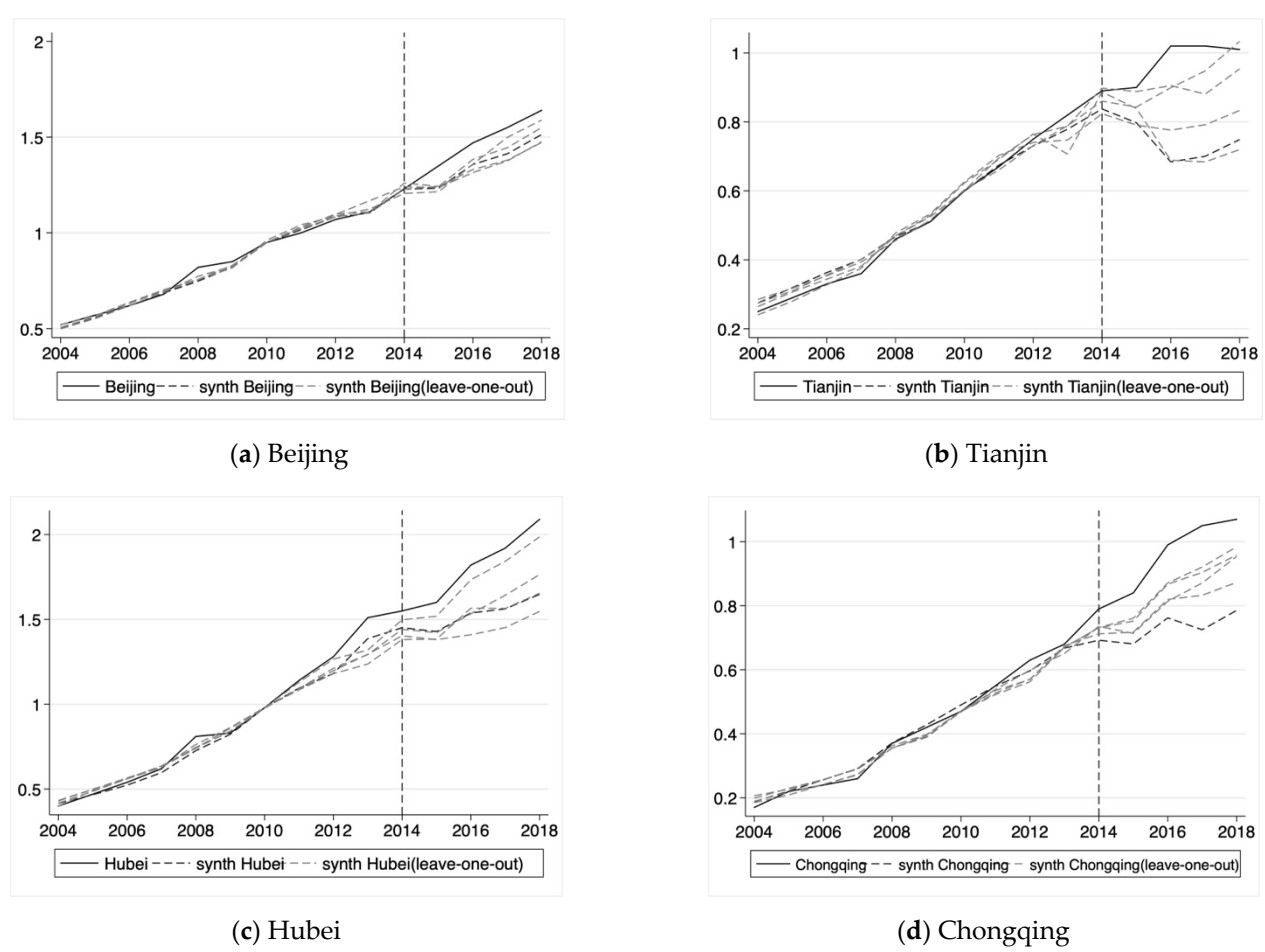

(**a**) Beijing

(**b**) Tianjin

(**c**) Hubei

(**d**) Chongqing

**Figure 6.** Results of the robustness test.

## 5. Analysis of Influence Mechanism

### 5.1. Results of Mediating Test

The above analysis shows that the carbon emission trading policies have a promotion effect on the green GDP of the pilot areas. Next, the policies transmission mechanism were studied through Formulas (7)~(11), and the results are shown in Table 4. It is not difficult to find the following.

Firstly, the direct effect of carbon emission trading policies on green GDP is significantly negative. The indirect effect is significantly positive, which is due to the fact that the carbon emission trading policies limit the excessive economic growth to a certain extent, resulting in a decline in GDP. However, the policies reduce the damage of economic development to the environment by encouraging enterprises to improve production technology, as well as adjusting and optimizing industrial structure and energy structure so that the total effect is still significantly positive. This result is consistent with the goal of promoting high-quality development of China's economy through carbon emission trading policies.

Secondly, the direct and indirect impacts of carbon emission trading policies on the level of technology, industrial structure, and energy structure are consistent. Specifically, the policies have significantly increased the *R&D* investment of industrial enterprises and the proportion of clean energy use. They have also promoted the transfer of the secondary industry to the tertiary industry. This result further proves that carbon emission trading

policies have a promotional effect on green GDP in the pilot areas, and there are many mechanisms contributing to this promotion.

**Table 4.** Test results of the mechanism of carbon emission trading policies to enhance green GDP.

| Variable | | Dependent Variable | | | |
|---|---|---|---|---|---|
| | | *Rd* | *Indus* | *Energy* | *Ggdp* |
| **Independent Variable** | | **(1)** | **(2)** | **(3)** | **(4)** |
| *Pilot × Year* | Direct | 0.64 *** (13.79) | −0.16 *** (−2.53) | 0.01 (1.07) | −0.39 *** (−2.49) |
| | Indirect | | −0.33 *** (−7.68) | 0.02 *** (3.93) | 1.22 *** (8.11) |
| | Overall | 0.64 *** (13.79) | −0.49 *** (−8.36) | 0.03 *** (3.99) | 0.84 *** (4.64) |
| *R&D* | Direct | | −0.52 *** (−9.25) | 0.01 ** (1.98) | 2.13 *** (13.14) |
| | Indirect | | | 0.01 *** (2.96) | −0.17 * (−1.91) |
| | Overall | | −0.52 *** (−9.25) | 0.02 *** (3.51) | 1.96 *** (11.58) |
| *Indus* | Direct | | | −0.02 *** (−3.13) | 0.63 *** (4.87) |
| | Indirect | | | | −0.12 *** (−2.86) |
| | Overall | | | −0.02 *** (−3.13) | 0.51 *** (3.79) |
| *Energy* | | | | | 6.99 *** (7.15) |

Note: *, ** and *** represent the level of significance of 10%, 5%, and 1%, respectively.

Thirdly, the direct effect of the level of technology on the green GDP is significantly positive, while the indirect effect is significantly negative. The direct effect comes from the *R&D* investment of industrial enterprises. Obviously, the higher the investment is, and the higher the cleaner production technology of enterprises becomes, the more the green GDP will improve. Indirect effects come from two aspects. From the perspective of industrial structure, improvement of the level of technology means that the *R&D* cost of industrial enterprises increases, which leads to a decline in profits of industrial enterprises and promotes the transfer of industrial structure from the secondary industry to the tertiary industry. This may lead to the decline of GDP. From the perspective of energy structure, the direct effect of improving the level of technology on energy structure is weak, but it can force the proportion of clean energy to increase through the impact on industrial structure, thus reducing energy consumption and loss. The net effect after the superposition of indirect effects is negative, which indicates that the total GDP decline caused by the change in industrial structure is more obvious, while the total effect after the superposition of direct effects and indirect effects is still significantly positive. This indicates that the improvement of the level of technology is an important means to achieve high-quality green economic development.

Fourthly, the direct effect of industrial structure on green GDP is significantly positive, which shows that industrial output is still an important part of GDP at present, and that the structural change from the secondary industry to the tertiary industry will lead to the decline of total GDP, thereby reducing green GDP. The indirect effect is significantly negative, and is achieved by circuitous influence on energy structure—it can be found that the impact of industrial structure on energy structure is significantly negative. However, the impact of

energy structure on green GDP is significantly positive. Thus, the structural change from the secondary industry to the tertiary industry will promote gradual replacement of fossil energy consumption by power consumption, especially with the increasing proportion of clean power such as wind and photovoltaic power. Its positive promotional effect on green GDP will become more obvious. Overall, the total effect of industrial structure on green GDP is significantly positive, and thus the direct effect is more significant than the indirect effect.

Finally, the impact of energy structure on green GDP is significantly positive, which has been explained above, so it will not be repeated here.

### 5.2. Test Results of Policies Transmission Path

Next, we further tested the chain mediating effect so as to more accurately observe the mutual influence effect among the mediation variables and to clarify the transmission path of policies. The results are shown in Table 5.

**Table 5.** Test results of chain mediating effect.

| Path | Chain Intermediary | Coefficient | 95% Confidence Interval | Is There a Mediating Effect |
|------|--------------------|-------------|-------------------------|-----------------------------|
| 1 | policies → Technical Level → Green GDP | 1.35 *** (9.55) | [1.07, 1.63] | Yes |
| 2 | policies → Industrial Structure → Green GDP | −0.10 *** (−2.73) | [−0.18, −0.03] | Yes |
| 3 | policies → Energy Structure → Green GDP | 0.06 (1.23) | [−0.03, 0.15] | No |
| 4 | policies → Technical Level → Industrial Structure → Green GDP | −0.21 *** (−4.81) | [−0.29, −0.12] | Yes |
| 5 | policies → Technical Level → Energy Structure → Green GDP | 0.06 (1.44) | [−0.02, 0.15] | No |
| 6 | policies → Industrial Structure → Energy Structure → Green GDP | 0.02 * (1.76) | [−0.002, 0.04] | Yes |
| 7 | policies → Technical Level → Industrial Structure → Energy Structure → Green GDP | 0.04 ** (2.46) | [0.01, 0.07] | Yes |

Note: *, ** and *** represent the level of significance of 10%, 5%, and 1%, respectively.

Table 5 shows that there are five main transmission paths of carbon emission trading policies to green GDP, which can be divided into major categories. The first category takes the technical level as the starting point, which includes three paths: Path 1, policies → technical level → green GDP, with a coefficient of 1.35; Path 4, policies → technical level → industrial structure → green GDP, with a coefficient of −0.21; Path 7, policies → technical level → industrial structure → energy structure → green GDP, with a coefficient of 0.04. The second category takes industrial structure as the starting point, which includes two paths: Path 2, policies → industrial structure → green GDP, with a coefficient of −0.10; Path 6, policies → industrial structure → energy structure → green GDP, with a coefficient of 0.02. Comparing the relative size and positivity and negativity among the above path coefficients, we can find the following.

Firstly, with the increasing number of circuitous transmission in the policies path, the effect of the influence of policies is gradually weakened.

Secondly, the coefficients of Path 2 and Path 4 are negative, and the common factor is that industrial structure is the end point of the policies' transmission path (Path 2 is also the starting point). This shows that the structural change from the secondary industry to the tertiary industry does not necessarily lead to the increase in green GDP, but the result may be just the opposite. Due to this structural change, the total GDP declines, which exceeds the incomes of environmental protection. The coefficients of Paths 1, 6, and 7 are positive,

which indicates that the growth of the green GDP can be truly realized by increasing the technical level and the proportion of clean energy.

Thirdly, by further comparing the absolute values of the coefficients of Paths 2, 4, 6, and 7, we can find that the effect of transmitting policies through the technical level is stronger than doing so through industrial structure, although the former is more circuitous. This means that the technical level is the source of improvement for the green GDP. Only when enterprises upgrade their own energy-saving and emission-reduction technologies and eliminate backward production capacity can they fundamentally realize green development of the economy.

Finally, although the coefficients of Paths 3 and 4 are not significant, they are both positive with those of Paths 6 and 7, and the common point is that the energy structure is the end of the policies' transmission paths. Therefore, the optimization of energy structure is an important foothold for the transmission path of carbon emission trading policies. Whether it is the direct impact of policies or the circuitous impact of the technical level and industrial structure, the increase in the proportion of clean energy is beneficial to the promotion of green GDP. This also means that the extensive use of clean energy is a necessary condition for China's economy to achieve green and high-quality development in the future.

## 6. Conclusions and Recommendations of Policies

Carbon emission trading has always been regarded as an important measure to promote the development of green economy. However, academic circles have not tested the impact of these policies on China's green economy in detail, and there is little discussion on its complex impact mechanism. Based on the panel data of 29 provinces in China from 2004 to 2018, this paper, making use of the pilot event of carbon emission trading initiated by the National Development and Reform Commission in 2013, empirically tests the impact of carbon emission trading policies on green economy in pilot areas using the synthetic control method. It also studies the mediating mechanism of the effects of the policies on the green economy. The study found that:

First, carbon emission trading policies have significantly improved the level of green economy in the pilot areas, and played a successful role in the demonstration of the developing green economy in China. However, due to the differences in economic foundation, industrial structure, and location conditions of the pilot projects, the promotional effects of the policies are also different. The results of the synthetic control test show that the policies have significantly improved the green GDPs of Beijing, Tianjin, Hubei, and Chongqing, among which the improvement rate of Beijing is relatively low, while those of Tianjin, Hubei, and Chongqing are relatively high. This result reflects the importance of reasonable institutional design for improving green GDP. Siminicg et al. [48] pointed out that EU countries which adopted public policies such as green procurement experienced a significant positive impact on the development of their green economies.

Second, the carbon emission trading policies mainly promote the green economy through two mediating mechanisms: the improvement of technical level and the adjustment of energy structure. The adjustment of industrial structure cannot independently promote the development of the green economy, but needs to complement the improvement of the technical level and the adjustment of the energy structure. This result is consistent with many current studies, that is, the development and application of renewable energy technologies have a positive impact on green GDP. For example, Sohag et al. [49] found that the use of renewable energy will achieve sustainable economic growth in Turkey. Taskin et al. [50] made similar conclusions in their research on renewable energy promoting green economic growth in OECD countries. It also shows that the China's government should pay more attention to the technical level and energy structure when promoting the development of the green economy, instead of mechanically promoting the transfer from the secondary industry to the tertiary industry as the foothold.

Based on this, this paper puts forward the following suggestions:

First, with the background of the launch of the national carbon emission trading market, we will continue to improve the market system and system construction, promote the implementation of emission reduction policies by means of the market, promote the green economy in all parts of the country, especially in the underdeveloped areas in the central and western regions, reduce the consumption and destruction of environmental resources caused by economic development, and realize the harmonious development between man and nature.

Second, cleaner production technology plays a vital intermediary role in the process of promoting the green economy by carbon emission trading policies. Therefore, in the process of implementing carbon emission reduction policies, we should give full attention to scientific and technological empowerment and closely watch the incremental changes brought by scientific and technological progress to the development of the green economy. The government's environmental regulation and environmental protection subsidy measures for enterprises should consider the development of cleaner production technology to be the breakthrough point, and provide necessary policies and financial support for enterprises in combination with the green credit of the financial system.

Third, the optimization of energy structure is conducive to playing a policies role, promoting the factors of green economy, and complementing the development of cleaner production technology. Therefore, the adjustment and upgrading of the industrial structure should consider the optimization of energy structure, and the development and application of clean energy such as hydro-power, wind energy, nuclear energy and solar energy, to be the national strategies for developing a green economy in China. Thus, we will promote the adjustment and upgrading of the industrial structure with the optimization of energy structure, and truly realize the harmonious coexistence of economic construction and environmental protection.

**Author Contributions:** Conceptualization, L.W.; Data curation, Z.C. and Z.H.; Formal analysis, Z.C.; Investigation, Z.C. All authors have read and agreed to the published version of the manuscript.

**Funding:** This research was funded by the Key Project of Research on carbon asset evaluation and management strategy of China Southern Power Grid Co. Ltd. grant number ZBKJXM20210238.

**Informed Consent Statement:** Informed consent was obtained from all subjects involved in the study.

**Data Availability Statement:** The data presented in this study are available on request from the corresponding author. The data are not publicly available due to privacy restrictions.

**Conflicts of Interest:** The authors declare no conflict of interest.

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
