# Peer review of "Research on the Effects and Mechanism of Carbon Emission Trading on the Development of Green Economy in China"

_sustainability, doi:10.3390/su141912483_

Round 1
Reviewer 1 Report
The article is devoted to an urgent topic - the formation and regulation of the carbon market. The article fully substantiates the research materials using the mathematical apparatus (the method of synthetic control and the test of the mediating effect). However, there are some comments:1) Since the study was conducted in China, it may be necessary to indicate in the title the Study of the Impact and Mechanism of Carbon Emissions Trading on the Development of the green economy in China (or on the example of the provinces of China).
2) Very small text of the signatures of the axes of the graphs in Figures 2-6, it is desirable to make it more readable.
3) In paragraph 3.5. Control variables, I would like to know why the authors use criterion (3) Number of medical personnel per capita: the impact of the number of medical personnel per capita on green GDP? What is the reason for using this particular criterion? Why not, for example, rank by population groups with different income levels or take into account the age groups of the disabled population, etc.? The relationship between green GDP and medical personnel is not very clear.
Author Response
Response to Reviewer 1 Comments
Point 1: Since the study was conducted in China, it may be necessary to indicate in the title the Study of the Impact and Mechanism of Carbon Emissions Trading on the Development of the green economy in China (or on the example of the provinces of China).
Response 1: Thank you very much for your advice. We will change the title to " Research on the Effects and Mechanism of Carbon Emission Trading on the Development of Green Economy in China"
Point 2: Very small text of the signatures of the axes of the graphs in Figures 2-6, it is desirable to make it more readable.
Response 2: We have adjusted the size of the signatures of the axes in Figure 2-6 to make it more readable.
Point 3: In paragraph 3.5. Control variables, I would like to know why the authors use criterion (3) Number of medical personnel per capita: the impact of the number of medical personnel per capita on green GDP? What is the reason for using this particular criterion? Why not, for example, rank by population groups with different income levels or take into account the age groups of the disabled population, etc.? The relationship between green GDP and medical personnel is not very clear.
Response 3: We hope to find an appropriate exogenous variable that can affect life expectancy in different regions, which affects the "income from environmental improvement" of green GDP. Therefore, we selected the number of medical personnel per capita as a control variable. In fact, we have considered using the income level as a control variable, but we have eliminated it because we are worried about the problem of taking the consequences as the cause. Thank you very much for your advice.

Reviewer 2 Report
original paper, but lacking introductory and methodological details.
it is necessary that the authors increase the definition of the circular economy and the green economy in the introductory part and that they better explain the reasons for the choice of the particular variables analyzed.
Results are clearly presented but comparisons with other international territorial realities are lacking
Author Response
Response to Reviewer 2 Comments
Point 1: original paper, but lacking introductory and methodological details.
it is necessary that the authors increase the definition of the circular economy and the green economy in the introductory part and that they better explain the reasons for the choice of the particular variables analyzed.
Results are clearly presented but comparisons with other international territorial realities are lacking
Response 1: Thank you very much for your advice. We added the definition of green economy and circular economy in Section 2:
Environmental economists believe that economic development must be sustainable for the natural and human beings, and should not lead to social division and ecological crisis due to blind pursuit of production growth. They advocated the establishment of an "affordable economy" based on social and ecological conditions. Therefore, the essence of green economy is a sustainable development economy with the coordinated development of ecology and economy as the core. It is an economic development mode characterized by maintaining the human living environment, reasonably protecting resources and benefiting human health. Green economy is essentially consistent with circular economy, which emphasizes resource conservation and recycling.
We also added some comparisons with international studies in Section 6:
First, the carbon emission trading policies have significantly improved the level of green economy in the pilot areas and played a successful demonstration role in developing green economy in China. However, due to the differences in economic foundation, industrial structure and location conditions of the pilot projects, the promotion effects of policies are also different. The results of synthetic control test show that the policies has significantly improved the green GDP of Beijing, Tianjin, Hubei and Chongqing, among which the improvement rate of Beijing is relatively low, while that of Tianjin, Hubei and Chongqing is relatively high. This result reflects the importance of reasonable institutional design for improving green GDP. Siminicg et al. (2020) pointed out that EU countries that adopted public policies such as green procurement had a significant positive impact on the development of green economy.
Second, the carbon emission trading policies mainly promote the green economy through two mediating mechanisms: the improvement of technical level and the adjustment of energy structure; The adjustment of industrial structure cannot independently promote the development of green economy, but needs to complement the improvement of technical level and the adjustment of energy structure. This result is consistent with many current studies, that is, the development and application of renewable energy technologies have a positive impact on green GDP. For example, Sohag et al. (2019) found that the use of renewable energy will achieve sustainable economic growth in Turkey; Taskin et al.(2020) made similar conclusions in their research on renewable energy promoting green economic growth in OECD countries.
